# Measurement-Based Control of Quantum Entanglement and Steering in a Distant Magnomechanical System

## Huatang Tan

Department of Physics, Huazhong Normal University, Wuhan 430079, China; tht@mail.ccnu.edu.cn

**Abstract:** In this paper, we propose a scheme for measurement-based control of hybrid Einstein–Podolsky–Rosen (EPR) entanglement and steering between distant macroscopic mechanical oscillator and yttrium iron garnet (YIG) sphere in a system of an electromechanical cavity unidirectionally coupled to an electromagnonical cavity. We reveal that when the output of the electromagnonical cavity is continuously monitored by homodyne detection, not only the phonon–magnon entanglement and steering but also the purities of the phononic, magnonic and phonon–magnon states are considerably enhanced. We also find that the measurement can effectively retrieve the magnon-to-phonon steering, which is not yet obtained in the absence of the measurement. We show that unconditional phonon–magnon entanglement and steering can be achieved by introducing indirect feedback to drive the magnon and mechanical subsystems. The long-distance macroscopic hybrid entanglement and steering can be useful for, e.g., fundamental tests for quantum mechanics and quantum networks.

**Keywords:** cavity magnonics; cavity optomechanics; magnomechanical entanglement; continuous weak measurement

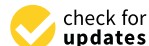

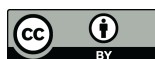

## 1. Introduction

In the past decade, great progress toward the realization of nonclassical phenomena in macroscopic mechanical oscillators has been achieved [1–4], motivated by its potential applications in the fields of probing the fundamental limits of quantum theory [5,6], ultrahigh precision measurements [7–10], and quantum information [11,12]. So far, various quantum effects [13–21], including mechanical squeezing [13], entanglement and Bell nonlocality [14–16], and phonon Fock states [20], have been demonstrated. Moreover, continuous-measurement-based control of mechanical oscillators has also been investigated experimentally [22–27], including cooling a mechanical resonator closely to ground states, observing quantum trajectory, and entropy production of a continuously monitored optomechanical oscillator. In addition, proposals for preparing strong mechanical squeezing, optomechanical entanglement and steering through continuous homodyne detection have been put forward [28,29].

In parallel with well-studied cavity optomechanical systems, macroscopic quantum effects in hybrid systems based on magnons in magnetic materials such as YIG sphere being around hundreds of micrometers in diameter increasingly have recently attracted a lot of attention, since magnons possess great frequency tunability, very low loss, and good coupling capability to other systems, i.e., microwave or optical photons, phonons, and qubits [30,31]. This therefore provides efficient ways to control quantum effects of magnons and meanwhile makes magnon-based systems very suitable for various quantum tasks. Qubit-based magnon quanta sensing, single-magnon control, and generation of a macroscopic Bell state between a single magnon and a superconducting qubit have already been reported [32–34]. Dynamical backaction magnomechanics and mechanical bistability have also been observed in cavity magnomechanics [35,36]. Recent proposals have been put forward for achieving magnon entanglement, magnon squeezing, magnon-mediated microwave photon entanglement, magnon blockade, magnon cat states [37–46]. In addition, magnon-based hybrid systems can also exhibit rich nonlinear phenomena, optomagnonic frequency combs [47], magnonic frequency combs in cavity magnomechanics [48], and magnon-induced high-order sideband generation [49].

In this paper, we consider a continuous measurement scheme to control photon-mediated EPR entanglement between a mechanical resonator and a distant YIG sphere. Such long-distance entanglement enables quantum communication among remote quantum nodes in quantum networks [50,51]. It should be noted that distant EPR entanglement between macroscopic mechanical and spin systems has been realized via unidirectional light coupling and measurement [52]. In addition, recent experiments have realized entanglement and quantum discord between superconducting qubits by virtue of continuous measurements, e.g., photon counting and homodyne detection [53,54]. We consider a system of an electromechanical cavity unidirectionally coupled to an electromagnonical cavity where a YIG is placed. It is found that when the output of the electromagnonical cavity is continuously monitored by homodyne detection, the phonon–magnon entanglement and steering can be obviously enhanced. We reveal that enhancement due to measurement enlarges the stability regime and also enhances the purities of the phononic, magnonic and phonon–magnon states. The measurement can also retrieve the steering from magnons to phonons which is unachievable in the absence of the measurement. We also show that by introducing indirect feedback to the mechanical and magnonic subsystems, unconditional phonon–magnon entanglement and steering can be achieved.

This paper is organized as follows. In Section 2, the system and working equations are presented. In Section 3, the effects of the continuous measurement on the phonon–magnon entanglement and steering are investigated in detail. Section 4, the indirect feedback is introduced to achieve unconditional entanglement and steering. In the last Section 5, the conclusion is given.

## 2. System and Equations

As shown in Figure 1, we consider that a driven electromechanical cavity is unidirectionally coupled to an electromagnonical cavity. For the electromechanical cavity, the cavity resonance is modulated by the mechanical oscillator, giving rise to the electromechanical coupling. While inside the electromagnonical cavity, a ferrimagnetic YIG sphere with a diameter of about hundreds of micrometers is placed and also biased in an uniform magnetic field. The magnons, characterizing the quanta of collective spin excitations in the YIG sphere, are coupled to the cavity mode via magnetic dipole interaction. In the rotating frame with respect to the driving frequency $\omega_d$, the system's Hamiltonian reads ($\hbar = 1$)

$$
\begin{aligned}
\hat{H} = &\sum_{j=1,2} \delta_j \hat{A}_j^\dagger \hat{A}_j + \delta_m \hat{M}^\dagger \hat{M} + \omega_b \hat{B}^\dagger \hat{B} \\
&+ \bar{g}_{ab} \hat{A}_1^\dagger \hat{A}_1 (\hat{B} + \hat{B}^\dagger) + g_{am}(\hat{A}_2^\dagger \hat{M} + \hat{A}_2 \hat{M}^\dagger) \\
&- i(\mathcal{E}_d^* \hat{A}_1 - \mathcal{E}_d \hat{A}_1^\dagger),
\end{aligned}
\tag{1}
$$

where the bosonic annihilation operators $\hat{A}_j$, $\hat{M}$ and $\hat{B}$ denote, respectively, the $j$th cavity, magnon (magnetostatic), and mechanical modes of resonances $\omega_{c_j}$, $\omega_m$ and $\omega_b$, respectively. The detunings $\delta_j = \omega_{c_j} - \omega_d$ and $\delta_m = \omega_m - \omega_d$. $\bar{g}_{ab}$ represent electromechanical coupling with a single microwave photon, and $g_{am}$ denotes electromagnonical coupling, dependent on the number of spins. For the strong drive with the amplitude $\mathcal{E}_d$, Equation (1) can be linearized around the steady-state amplitudes by replacing the operators by $\hat{A}_j = \langle \hat{A}_j \rangle_{\mathrm{ss}} + \hat{a}_j$, $\hat{B} = \langle \hat{B} \rangle_{\mathrm{ss}} + \hat{b}$, and $\hat{M} = \langle \hat{M} \rangle_{\mathrm{ss}} + \hat{m}$, and the resulted linear Hamiltonian is given by

$$
\begin{aligned}
\hat{H}_{\mathrm{lin}} = &\Delta_1 \hat{a}_1^\dagger \hat{a}_1 + \delta_2 \hat{a}_2^\dagger \hat{a}_2 + \delta_m \hat{m}^\dagger \hat{m} + \omega_b \hat{b}^\dagger \hat{b} \\
&+ g_{ab}(\hat{a}_1 + \hat{a}_1^\dagger)(\hat{b} + \hat{b}^\dagger) + g_{am}(\hat{a}_2^\dagger \hat{m} + \hat{a}_2 \hat{m}^\dagger),
\end{aligned}
\tag{2}
$$

with $\Delta_1 = \delta_1 + 2\bar{g}_{ab}\mathrm{Re}[\langle \hat{B} \rangle_{\mathrm{ss}}]$ and $g_{ab} = \bar{g}_{ab}\langle \hat{A}_1 \rangle_{\mathrm{ss}}$.

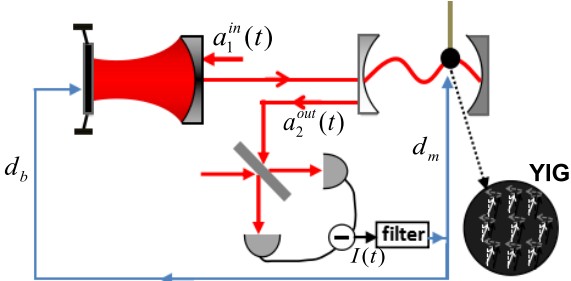

**Figure 1.** The schematic plot of a driven electromechanical cavity is unidirectionally coupled to an electromagnonical cavity whose output is continuously monitored by homodyne detection. Based on the detection outcome $I(t)$, indirect (state-based) feedback with gains $d_{b,m}$ is employed to achieve unconditional entanglement and steering between the macroscopic mechanical oscillator and the YIG sphere.

We further consider that the output field $\hat{a}_2^{\text{out}}(t)$ of the second cavity $\hat{a}_2$ is subject to time-continuous homodyne detection. For the cascade two-cavity system, the input–output relation is

$$\hat{a}_2^{\text{out}}(t) = \sqrt{\kappa_2}\hat{a}_2 + \sqrt{\xi_e\kappa_1}\hat{a}_1 + \sqrt{\xi_e}a_1^{\text{in}}(t), \tag{3}$$

where $\hat{a}_1^{\text{in}}(t)$ is the input of the first cavity $\hat{a}_1$ satisfying $\langle\hat{a}_1^{\text{in}}(t)\hat{a}_1^{\text{in}\dagger}(t')\rangle = \delta(t-t')$, $\kappa_j$ are the cavity dissipation rates, and $\xi_e$ accounts for the cascade–cavity coupling efficiency. When the generalized quadrature,

$$\hat{x}_{2\theta}^{\text{out}} = (\hat{a}_2^{\text{out}}e^{i\theta} + \hat{a}_2^{\text{out}\dagger}e^{-i\theta})/\sqrt{2}, \tag{4}$$

is homodynely detected with the local phase $\theta$, the detection current

$$I_\theta dt = \sqrt{\eta}\big\langle(\sqrt{\xi_e\kappa_1}\hat{a}_1 + \sqrt{\kappa_2}\hat{a}_2)e^{i\theta} + H.c.\big\rangle dt + dW. \tag{5}$$

Here, $\eta$ denotes the detection efficiency and $dW$ is the stochastic Wiener increment satisfying the Ito rule $(dW)^2 = dt$. Conditioned on the detection outcomes, the stochastic master equation for the density operator $\hat{\rho}_s^c$ of the whole system is given in [55,56]:

$$
\begin{aligned}
d\hat{\rho}_s^c = {} & -i\big[\hat{H}_{\text{lin}}, \hat{\rho}_s^c\big]dt + \kappa_1\mathcal{L}[\hat{a}_1]\hat{\rho}_s^c dt + \kappa_2\mathcal{L}[\hat{a}_2]\hat{\rho}_s^c dt \\
& + \sum_{j=m,b}\kappa_j\Big\{(\bar{n}_j^{\text{th}}+1)\mathcal{L}[\hat{j}]\hat{\rho}_s^c dt + \bar{n}_j^{\text{th}}\mathcal{L}[\hat{j}^\dagger]\hat{\rho}_s^c dt\Big\} \\
& - \sqrt{\xi_e\kappa_1\kappa_2}\Big([\hat{a}_2^\dagger, \hat{a}_1\hat{\rho}_s^c] + [\hat{\rho}_s^c\hat{a}_1^\dagger, \hat{a}_2]\Big)dt, \\
& + \sqrt{\eta}\mathcal{H}\big[(\sqrt{\xi_e\kappa_1}\hat{a}_1 + \sqrt{\kappa_2}\hat{a}_2)e^{i\theta}\big]\hat{\rho}_s^c dW,
\end{aligned} \tag{6}
$$

where the symbols $\mathcal{L}[\hat{o}]\hat{\rho} = \hat{o}\hat{\rho}\hat{o}^\dagger - \frac{1}{2}(\hat{o}^\dagger\hat{o}\hat{\rho} + \hat{\rho}\hat{o}^\dagger\hat{o})$ and $\mathcal{H}[\hat{o}]\hat{\rho} = \hat{o}\hat{\rho} + \hat{\rho}\hat{o}^\dagger - \langle\hat{o} + \hat{o}^\dagger\rangle$. The second and third terms in the first line in Equation (6) describe the cavity dissipation processes, the second line describes the damping of the mechanical and magnon modes at the rates $\kappa_b$ and $\kappa_m$, in thermal environments with the mean thermal excitation numbers $\bar{n}_j^{\text{th}} \equiv (e^{\hbar\omega_j/k_B T_j} - 1)^{-1}$, for temperature $T_j$ and the Boltzmann constant $k_B$; the third line describes the cascade–cavity coupling with the efficiency $\xi_e$; and the last line characterizes the backaction due to the continuous measurement and it disappears when ensemble average is performed.

For initial Gaussian states, the system governed by Equation (6) evolves still in Gaussian, determined by the covariance matrix $\sigma_{c,jj'} = \langle\mu_j\mu_{j'} + \mu_{j'}\mu_j\rangle/2 - \langle\mu_j\rangle\langle\mu_{j'}\rangle$, where $\mu = (\hat{x}_1, \hat{p}_1, \hat{x}_2, \hat{p}_2, \hat{x}_b, \hat{p}_b, \hat{x}_m, \hat{p}_m)$ for the quadrature operators $\hat{x} = (\hat{o} + \hat{o}^\dagger)/\sqrt{2}$ and $\hat{p} = -i(\hat{o} - \hat{o}^\dagger)/\sqrt{2}$. From Equation (6), we have

$$d\bar{\mu}^T = A\bar{\mu}^T dt + (\sigma_c C - \Gamma)dW, \tag{7}$$

$$\dot{\sigma}_c = A\sigma_c + \sigma_c A^T + D - (\sigma_c C - \Gamma)(\sigma_c C - \Gamma)^T, \tag{8}$$

where $\bar{\mu}_j \equiv \langle \mu_j \rangle$. The matrix

$$A = \begin{pmatrix} A_1 & 0 & A_{ab} & 0 \\ A_{12} & A_2 & 0 & A_{am} \\ A_{ab} & 0 & A_b & 0 \\ 0 & A_{am} & 0 & A_m \end{pmatrix}, \tag{9}$$

where $A_{x=\{1,2,b,m\}} = -\begin{pmatrix} \kappa_x & -2\Delta_x \\ 2\Delta_x & \kappa_x \end{pmatrix}/2$, with $\Delta_2 = \delta_2$, $\Delta_m = \delta_m$ and $\Delta_b = \omega_b$, $A_{ab} = -\begin{pmatrix} 0 & 0 \\ 0 & 2G_{ab} \end{pmatrix}$, $A_{am} = \begin{pmatrix} 0 & g_{am} \\ -g_{am} & 0 \end{pmatrix}$, and $A_{12} = -\sqrt{\xi_e \kappa_1 \kappa_2}I$. The matrix

$$D = \begin{pmatrix} D_1 & D_{12} \\ D_{12} & D_2 \end{pmatrix} \oplus \begin{pmatrix} D_b & 0 \\ 0 & D_m \end{pmatrix}, \tag{10}$$

where $D_j = \kappa_j I/2$, $D_{b,m} = \kappa_{b,m}(\bar{n}_{b,m}^{\text{th}} + 1/2)I$, and $D_{12} = \sqrt{\xi_e \kappa_1 \kappa_2}I/2$. The vectors

$$\begin{aligned} C^T = & \sqrt{2\eta}\left(\sqrt{\xi_e \kappa_1}\cos\theta, -\sqrt{\xi_e \kappa_1}\sin\theta, \sqrt{\kappa_2}\cos\theta, \right. \\ & \left. -\sqrt{\kappa_2}\sin\theta, 0, 0, 0, 0\right), \end{aligned} \tag{11}$$

$$\begin{aligned} \Gamma^T = & \sqrt{\frac{\eta}{2}}\left(\sqrt{\xi_e \kappa_1}\cos\theta, -\sqrt{\xi_e \kappa_1}\sin\theta, \sqrt{\kappa_2}\cos\theta, \right. \\ & \left. -\sqrt{\kappa_2}\sin\theta, 0, 0, 0, 0\right). \end{aligned} \tag{12}$$

We see from Equation (2) that the first moments are related to the detection outcomes and are thus stochastic. Nevertheless, these stochastic moments are independent of the entanglement and steering for the Gaussian states and can be cancelled out by introducing feedback, as is shown later. On the contrary, the covariance matrix $\sigma_c$ is independent of the outcomes and thus deterministic and it completely determines quantum statistics of the system. The effect of continuous measurement is embodied by the last nonlinear term of Equation (8) (originating from the last term of Equation (6)), which is crucial for achieving strong EPR entanglement and steering for the present scheme.

By solving Equation (8), the covariance matrix $\sigma_{bm}$ of the mechanical and magnonic system can be obtained and expressed in the form $\sigma_{bm} = \begin{pmatrix} \sigma_b & \sigma_{bm} \\ \sigma_{bm}^T & \sigma_m \end{pmatrix}$. The phonon–magnon entanglement can be quantified by the logarithmic negativity [57],

$$E_{bm} = \max\left[0, -\ln(2e)\right], \tag{13}$$

where $e = 2^{-1/2}\sqrt{\Sigma - \sqrt{\Sigma^2 - 4\det\sigma_{bm}}}$ and $\Sigma = \det\sigma_c + \det\sigma_m - 2\det c_{bm}$. The steering from phonons to magnons can be quantified by [58]

$$S_{m|b} = \max\left[0, \frac{1}{2}\ln\left(\frac{\det\sigma_b}{4\det\sigma_{bm}}\right)\right], \tag{14}$$

and, similarly, the reverse steering from magnons to phonons is determined by

$$S_{b|m} = \max\left[0, \frac{1}{2}\ln\left(\frac{\det\sigma_m}{4\det\sigma_{bm}}\right)\right]. \tag{15}$$

## 3. Conditional Entanglement and Steering via Measurement

In Figure 2, the phonon–magnon entanglement $E_{bm}$ in the steady-state regime is plotted for the two cases that the measurement is absent [(a1)–(c1), $\eta = 0$] and is present [(a2)–(c2), $\eta = 1$]. As shown in Figure 2(a1), we see that without the measurement, the

maximal entanglement occurs for the detuning $\Delta_m = -\omega_b$ and also for the bad cavity $\kappa_{1,2} \gg \{\omega_b, g_{ab}, g_{am}\}$ [59]. This is because the phonon–magnon entanglement roots from the effective parametric amplification interaction between the mechanical and magnon modes result from the cascade photon coupling, which is resonant at this detuning; their strength is proportional to the product of cavity dissipation rates $\kappa_{1,2}$. When the continuous monitoring is turned on, the entanglement appears over a wider range of detuning $\Delta_m$ and its maximal value at $\Delta_m = -\omega_b$ is obviously enhanced for the given cavity dissipation rate $\kappa = \kappa_j \in [0.5\omega_b, 15\omega_b]$. From Figure 2(b1), we see that without the measurement, the entanglement is present at steady-state regime $\Delta_1 \geq 0$ and becomes maximal in the vicinity of instability threshold $\Delta_1 = 0$ of the optomechanical system. However, as depicted in Figure 2(b2), the continuous measurement enlarges the stable regime of the optomechanical system to blue-detuned regime $\Delta_1 < 0$, even with mediate optomechanical coupling $g_{ab}$. As a matter of fact, this blue-detuned regime is conducive to the generation of photon–phonon entanglement, although it is prohibited by the stability for a generic optomechanical system. Since the phonon–magnon entanglement actually results from the distribution of the photon–phonon entanglement via the unidirectional cavity coupling, the stronger steady-state phonon–magnon entanglement is thus achievable at $\Delta_1 < 0$, with the help of the measurement. In addition, we can also see that apart from the blue-detuning regime, the entanglement is also obviously enhanced in red-detuned region $\Delta_1 > 0$. This is because the continuous measurement effectively reduces the fluctuations of the mechanical and magnon modes in this regime. Similarly, as shown in Figure 2(c1–c3), the entanglement is also improved by the measurement for different values of the couplings, $g_{ab}$ and $g_{am}$. We can see that in the presence of the measurement, stronger entanglement can be achieved even for smaller coupling $g_{am}$.

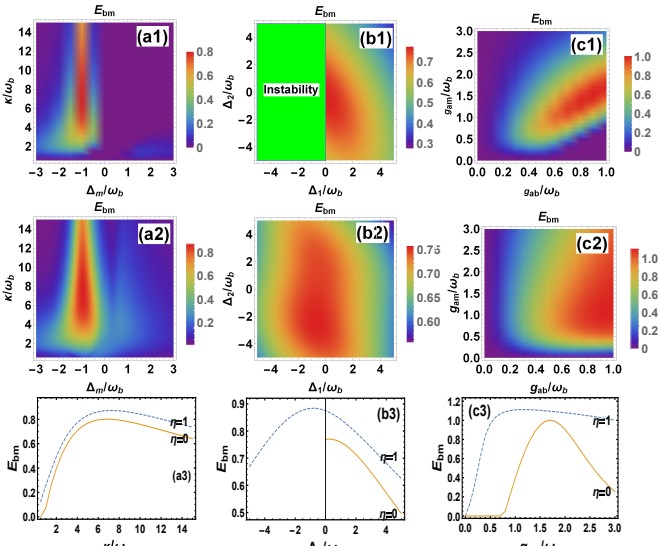

**Figure 2.** The dependence of the phonon–magnon entanglement $E_{bm}$ on different parameters for the measurement being absent ($\eta = 0$) in (**a1,b1,c1**) and present ($\eta = 1$) in (**a2,b2,c2**). In (**a1,a2**), $\Delta_1 = -\Delta_2 = \omega_b$, $g_{ab} = 0.5g_{am} = 0.5\omega_b$; In (**b1,b2**), $\kappa = 10\omega_b$, $\Delta_m = -\omega_b$, $g_{ab} = 0.5g_{am} = 0.5\omega_b$; In (**c1,c2**), $\kappa = 10\omega_b$, $\Delta_m = -\omega_b$, $\Delta_1 = -\Delta_2 = \omega_b$. The other parameters $\omega_b/2\pi = 10$ MHz, $\omega_m/2\pi = 10$ GHz, $\kappa_m = 0.15\omega_b$, $\kappa_b = 10^{-5}\omega_b$, $T_1 = T_m = T_b = 30$ mK, and $\xi_e = 1$. The plots (**a3,b2,c3**) are the same as (**a1,b1,c1**), respectively, with $\Delta_m = -\omega_m$ [in (**a3**)], $\Delta_2 = -\omega_m$ [in (**b3**)], and $g_{ab} = \omega_m$ [in (**c3**)].

In Figures 3 and 4, the steady-state steerings $S_{m|b}$ and $S_{b|m}$ are plotted, respectively. Comparing Figures 2 and 3, we can see that the properties of steering $S_{m|b}$ are similar to those of entanglement $E_{bm}$. On the contrary, steering $S_{b|m}$ displays quite different behaviors. As shown in plots (a1)–(b1) in Figures 3 and 4, when the measurement is absent, the steering from magnons to phonons is unachievable, i.e., $S_{b|m} = 0$, and thus, one-way steering ($S_{m|b} \neq 0$) is obtained in a wide range of parameters. This is mainly due to

dissipation rate $\kappa_b \ll \kappa_m$, which can be seen from Figure 5, where the steerings are plotted for decreased magnon damping rate $\kappa_m$. It can be found that the magnon-to-phonon steering appears and the degrees of the entanglement and steerings increases with the decreasing of the magnon damping. When the measurement is present, as depicted in Figure 4(a2,b2), the magnon-to-phonon steering also appears, and therefore the two-way steering between phonons and magnons can be obtained via the continuous measurement. This is also due to the fact that continuous measurement suppresses fluctuations of the magnon mode. Similarly to phonon-to-magnon steering $S_{m|b}$, the maximum of the reverse steering still occurs at $\Delta_m = -\omega_m$. From Figure 4(c1–c3), we see that in the presence of the continuous measurement, steering $S_{b|m}$ can obviously be enhanced and also exist over a wider range of coupling $g_{am}$.

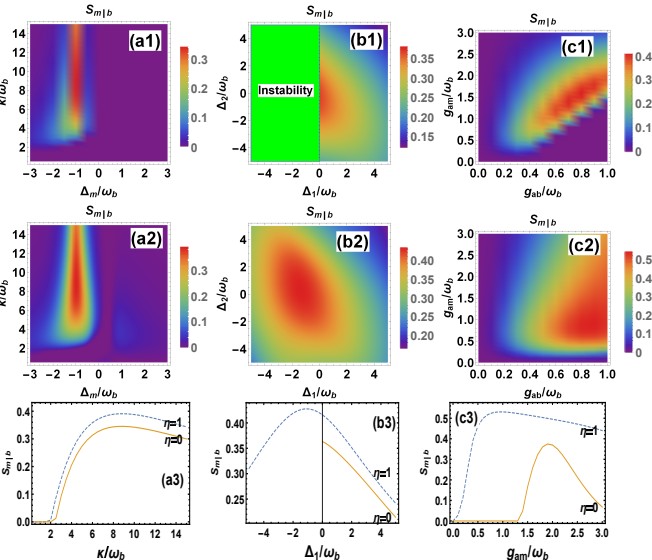

**Figure 3.** The dependence of phonon-to-magnon steering $S_{m|b}$ on different parameters for the measurement being absent ($\eta = 0$) (in (**a1,b1,c1**)) and present ($\eta = 1$) (in (**a2,b2,c2**)). The other parameters are the same as in Figure 2.

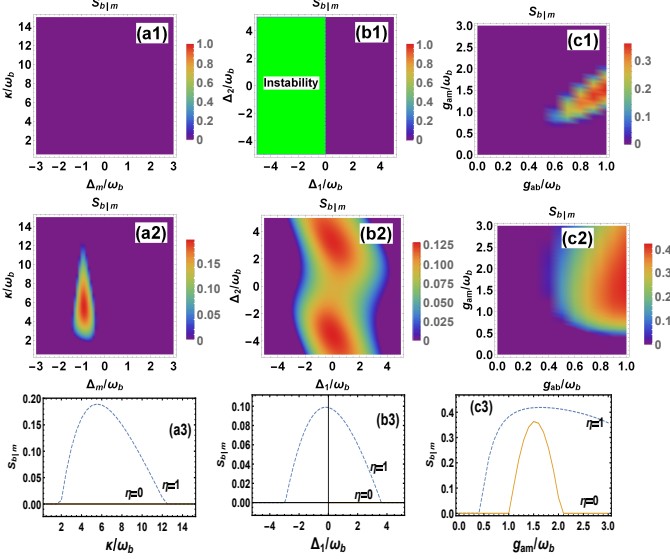

**Figure 4.** The dependence of magnon-to-phonon steering $S_{b|m}$ on different parameters for the measurement being absent ($\eta = 0$) (in (**a1,b1,c1**)) and present ($\eta = 1$) (in (**a2,b2,c2**)). The other parameters are the same as in Figure 2.

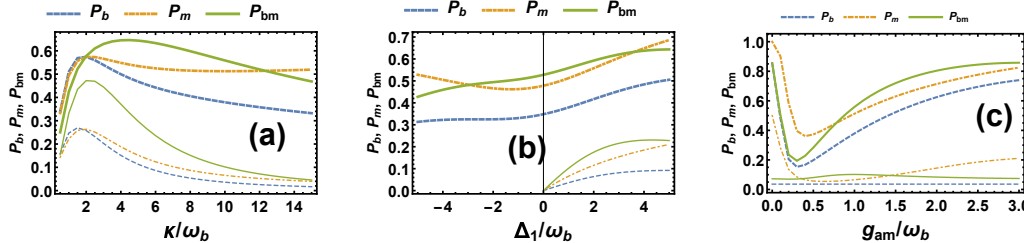

**Figure 5.** Purity $P_b$, $P_m$, and $P_{bm}$ of the phononic states, magnonic states, and phonon–magnon states in (**a**,**b**,**c**) for the same parameters as those in Figure 2(**a3**,**b3**,**c3**), for the measurement being present ($\eta = 1$, thick lines) and absent ($\eta = 0$, thin lines).

In Figure 6, we plot purity $P_b$, $P_m$, and $P_{bm}$ of the phononic states, magnonic states, and the phonon–magnon states for the cases that the measurement is present ($\eta = 1$) and absent ($\eta = 0$). It can be found that the purities are considerably enhanced by the measurement. Therefore, we can also understand that the measurement reduces the noise of the system and therefore enhances the entanglement and steering in the steady-state regime.

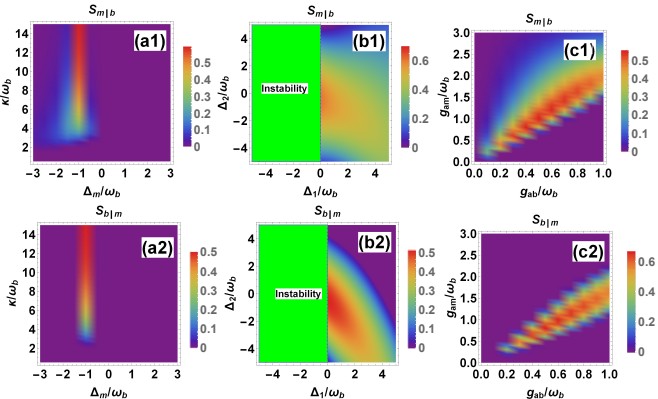

**Figure 6.** Density plots of steerings $S_{m|b}$ (**a1**–**c1**) and $S_{b|m}$ (**a2**–**c2**) in the absence of the measurement for the same parameters as (**a1**–**c1**) in Figure 3 and (**a1**–**c1**) in Figure 4, respectively, except for $\kappa_m = 0.01\omega_b$.

Figure 7 depicts the effects of imperfect unidirectional cavity coupling ($\xi_e < 1$) and thermal fluctuations on the entanglement and steering. It is shown that the entanglement and steering are robust against the unidirectional coupling loss. We see that $E_{bm}$ and $S_{m|b}$ can exist even for $\xi_e = 0.2$, whereas $S_{b|m}$ almost disappears for $\xi_e \leq 0.8$. For the current experiments in the microwave domain, unidirectional cavity coupling efficiency $\xi_e \approx 0.75$ is achieved [53,54]. In addition, we can also see that the entanglement can still be achieved even for the temperature of up to $T_b \approx 400$ mK. The steerings are more influenced by the thermal noise, and steering $S_{m|b}$ is more robust than the reverse steering, due to damping rates $\kappa_b \ll \kappa_m$. The two-way steering survives for $T \approx 100$ mK. For the YIG sphere, the cooling to 10 mK $\sim$ 1 K by using a dilution refrigerator, just with small line broadening, is achieved in experiment [60]. We note that around temperature $T_b = T_m \approx 30$ mK by using cryostat in experiments [60–63], the mean thermal magnon number $\bar{n}_m^{\text{th}} \approx 0$; it can thus be neglected, while the mean thermal phonon number $\bar{n}_b^{\text{th}} \approx 62$, not near the ground states. Here, the parameters can be chosen as $\omega_m/2\pi = 10$ GHz, $\omega_b/2\pi = 10$ MHz, electrome-chanical coupling $g_{ab}/2\pi = 5$ MHz for single-photon-mechanical coupling $\tilde{g}_{ab} \approx 150$ Hz and pump power $P_d \approx 1$ μW for $\Delta_1 = -\Delta_2 = -\Delta_m = \omega_m$ and $\kappa = 10\omega_m$. Experimentally, ultrastrong electromechanical coupling inducing the frequency splitting of 90 percent of bare 10-MHz mechanical frequency has already been achieved [61]. When considering a 400 μm diameter YIG sphere, electromagnonical coupling $g_{am} \equiv \sqrt{N}g_{m0} = \omega_b$ can be obtained for single-spin coupling $g_{m0}/2\pi \approx 38$ mHz [60] and the number of spins in sphere $N \approx 7 \times 10^{16}$ with net spin density of the sphere $\rho_s = 2.1 \times 10^{27}$ m$^{-3}$. For the present scheme, the optomechanical device can be a three-dimensional superconducting cavity

which is coupled to the motion of a micromechanical membrane. The electromagnonical device could also be a three-dimensional microwave cavity where an undoped single-crystal YIG sphere is coupled the the microwave field. The output ports of the two cavities can be connected via two microwave circulators and a coaxial cable to achieve the directional coupling from the electromechanical cavity to the electromagnonical cavity, as accepted in [53] where the second output is under continuous measurement. In the microwave domain, the cascade–cavity coupling efficiency of up to 0.8 can be achieved.

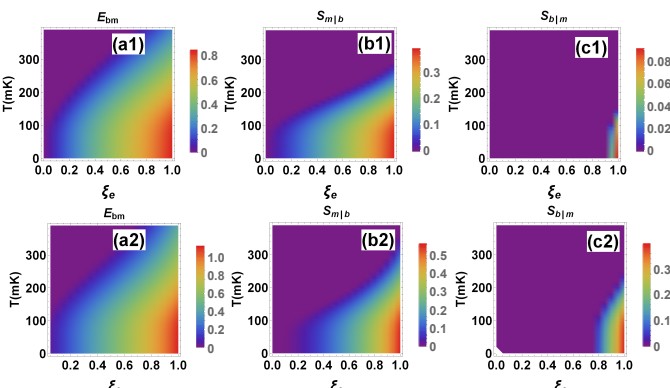

**Figure 7.** Density plot of entanglement $E_{bm}$ and steering $S_{m|b}$ and $S_{b|m}$ versus temperature $T$ and unidirectional coupling efficiency $\xi_e$ for $g_{ab} = 0.5\omega_b$ and $\Delta_1 = \omega_b$ in (**a1,b1,c1**), $g_{ab} = \omega_b$ and $\Delta_1 = -\omega_b$ in (**a2,b2,c2**), $g_{am} = \omega_b$, $\kappa = 10\omega_b$, $\Delta_2 = -\omega_b$, $\Delta_m = -\omega_b$, and the other parameters are the same as in Figure 2.

## 4. Unconditional Entanglement and Steering via Indirect Feedback

The above results are conditional, since expectation values $\bar{\mu}(t)$ are driven by stochastic noise $dW$ and therefore walk randomly in the phase space, depending on the detection outcomes. For many experimental runs of the system, the incoherent noise resulting from the random walk masks the conditional mechanical magnon EPR entanglement and steering studied in the above when ensemble average is performed. Therefore, to verify and apply the entanglement and steering, we need to convert the conditional results into the unconditional ones. This can be realized by employing state-based (indirect) feedback to remove the negative effect of stochastic expectation values $\bar{\mu}(t)$ [55,56]. Once the measurement is performed at some time $t$, values $\bar{x}_j(t)$ and $\bar{p}_j(t)$ can be inferred immediately, based on which the Markovian feedback described by the Hamiltonian

$$\hat{H}_{\text{fb}} = \sum_{j=b,m} d_p^j \bar{p}_j(t) \hat{x}_j - d_x^j \bar{x}_j(t) \hat{p}_j \tag{16}$$

can be constructed, with the feedback gain parameters $d_{x,p}^j$. The feedback leads Equation (7) to be modified by substituting $A$ with $\tilde{A} \equiv A + diag(0, 0, 0, 0, d_b^x, d_b^y, d_m^x, d_m^y)$.

For the feedback described by the Hamiltonian of Equation (16), the mechanical driving can be realized by, e.g., electric actuation, as demonstrated in [64–66]. To achieve the magnon damping force, a drive tone at frequency $\omega_d$ and supplied by a microwave source can directly drive the YIG sphere, as performed in [63]. Here, the driving magnetic field, bias magnetic field $H_B$, and the magnetic field of the microwave cavity are orthogonal to each other at the site of the YIG sphere to avoid the mutual impact among them.

Then, ensemble average $\bar{\sigma}_e \equiv \frac{1}{2}\langle \bar{\mu}_i(t)\bar{\mu}_j(t) + \bar{\mu}_j(t)\bar{\mu}_i(t)\rangle_e$ over many realizations of the system can be derived as

$$\dot{\bar{\sigma}}_e = \tilde{A}\bar{\sigma}_e + \bar{\sigma}_e\tilde{A}^T + (\sigma_c C - \Gamma)(\sigma_c C - \Gamma)^T, \tag{17}$$

as well as ensemble average correlation matrix

$$\sigma_e = \sigma_c + \bar{\sigma}_e \tag{18}$$

determining the system's properties under the feedback. When $\bar{\sigma}_e \approx 0$ through the feedback, the covariance matrix

$$\sigma_e \approx \sigma_c, \tag{19}$$

independent of the measurement outcomes and deterministic. The overlap between the states with the covariance matrices $\sigma_c$ and $\sigma_e$ can be quantified by fidelity [67]

$$\mathcal{F}_{\sigma_c,\sigma_e} = \left( \sqrt{\Theta} + \sqrt{\Lambda} - \sqrt{(\sqrt{\Theta} + \sqrt{\Lambda})^2 - \Delta} \right)^{-1}, \tag{20}$$

where $\Theta = 16 Det[J\sigma_c J\sigma_e + I/4]$, $\Lambda = 16 Det[\sigma_c + iJ/4] Det[\sigma_e + iJ/4]$, $\Delta = Det[\sigma_c + \sigma_e]$, and $J = \begin{pmatrix} 0 & 1 \\ -1 & 0 \end{pmatrix}$.

In Figure 8a,b, fidelity $\mathcal{F}_{\sigma_c,\sigma_e}$, entanglement $E_{bm}$ and steering $S_{m|b}$ and $S_{b|m}$ are plotted for the two cases: $d_x^j = d_p^j = d$ $(j = b, m)$ and $d_x^b = 0$ and $d_p^b = d_x^m = d_p^m = d$. We see that in both cases, the fidelity, the entanglement and steering increase as feedback strength $d$ arises. This is because the increase in the feedback strength leads to stronger damping for mean values $\langle \hat{x}_j \rangle$ and $\langle \hat{p}_j \rangle$, which in turn further suppress the fluctuations (i.e., $\bar{\sigma}_e$) of the mean values and even almost removes them completely in the limit of strong feedback. We see that in this limit, fidelity $F_{\sigma_c,\sigma_e} \to 1$ and the entanglement and steeering recover to the conditional values, $E_{bm} \to 0.84$, $S_{m|b} \to 0.39$, and $S_{b|m} \to 0.09$ in Figure 3 when $d_x^j = d_p^j = d$ and $d \gg \omega_b$. It is also explicitly shown that a greater amount of unconditional entanglement and steerings can be recovered when simultaneously driving mechanical position $\hat{x}_b$ and momentum $\hat{p}_b$ in the feedback compared to when just driving the mechanical position alone. In addition, as shown in Figure 8c,d, the effect of the unequal feedback parameters is taken into account. We see that the unconditional entanglement and steering become maximal approximately near the symmetric feedback and they decrease when $d_m$ exceeds $d_b$. For the asymmetric feedback, one-way steering also appears, showing that phonon–magnon entanglement and steering can be controlled via feedback.

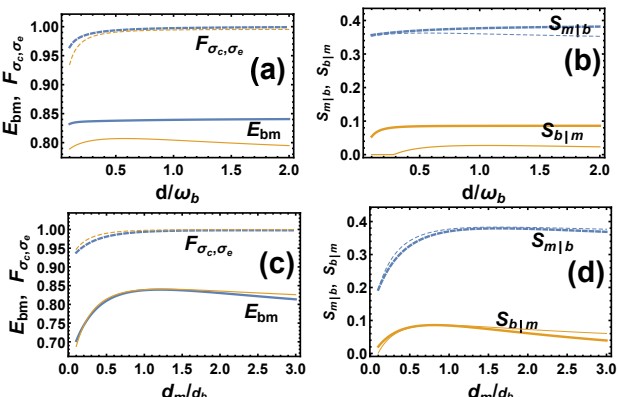

**Figure 8.** (**a**,**b**) The dependence of unconditional entanglement $E_{bm}$, fidelity $F_{\sigma_c,\sigma_e}$ and steerings $S_{m|b}$ and $S_{b|m}$ on feedback parameter $d$, respectively, for $d_x^j = d_p^j = d$ $(j = b, m)$ (thick lines) and $d_x^b = 0$ and $d_p^b = d_x^m = d_p^m = d$ (thin lines). (**c**,**d**) The effect of unequal feedback parameters $d_b$ and $d_m$ on entanglement and steerings for $d_b = 0.5\omega_b$ (thin lines) and $d_b = \omega_b$ (thick lines). We have $\Delta_1 = -\Delta_2 = \omega_b$, $g_{ab} = 0.5g_{am} = 0.5\omega_b$, $\kappa = 10\omega_b$, $\Delta_m = -\omega_b$, $\eta = 1$, $\xi_e = 1$, and the other parameters are the same as in Figure 2.

## 5. Conclusions

In conclusion, in this paper, we consider a continuous measurement scheme to enhance photon-mediated EPR entanglement and steering between a mechanical resonator and a distant YIG sphere, a system of an electromechanical cavity unidirectionally coupled to an electromagnonical cavity where a YIG is placed. We reveal that when the output of the electromagnonical cavity is continuously monitored by homodyne detection, the phonon–magnon entanglement and steering can be obviously enhanced. It is also revealed that the

enhancement due to the measurement enlarges the stability regime and also enhances the purities of the phononic, magnonic and phonon–magnon states. It is found that continuous measurement can retrieve the steering from magnons to phonons, which is unachievable in the absence of the measurement. We finally propose an indirect feedback scheme to achieve unconditional phonon–magnon entanglement and steering. The long-distance macroscopic hybrid entanglement and steering can be useful for, e.g., fundamental tests for quantum mechanics and quantum networks.

**Funding:** This work is supported by the National Natural Science Foundation of China (No.12174140).

**Institutional Review Board Statement:** Not applicable.

**Informed Consent Statement:** Not applicable.

**Data Availability Statement:** Not applicable.

**Conflicts of Interest:** The authors declare no conflict of interest.

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
