# Peer review of "Measurement-Based Control of Quantum Entanglement and Steering in a Distant Magnomechanical System"

_photonics, doi:10.3390/photonics10101081_

Round 1

Reviewer 1 Report

The authors propose a scheme for measurement-based control of EPR entanglement and steering between distant macroscopic mechanical oscillator and yttrium iron garnet (YIG) sphere in a hybrid system. In this work, the authors find that entanglement or steering can be improved by continuous detection or indirect feedback. The topic is interesting, results are reliable, and the manuscript is well organized and written. I believe the manuscript is suitable for acceptance by photonics after addressing the following one question, that is,

I notice that the cavity electromechanical subsystem is decoupled from the cavity-magnon subsystem in Eq. (1). But in the manuscript the authors claim that the hybrid entanglement such as phonon-magnon or steering can be achieved via continuous detection. So I guess that the measurement may introduce a coupling mechanism between this two decoupled subsystems. I would like to invite authors to revisit the manuscript and give an effective Hamiltonian to describe the mechanism of entanglement generation.

Reviewer 2 Report

In the manuscript "Measurement-based control of quantum entanglement and steering in a distant magnomechanical system", a scheme for measurement-based control of hybrid EPR entanglement and steering between distant macroscopic mechanical oscillator and YIG sphere is theoretically proposed. The author reveals that when the output of the electromagnonical cavity is continuously monitored by homodyne detection, the phonon-magnon entanglement and steering can be enhanced, and an indirect feedback scheme to achieve unconditional phonon-magnon entanglement and steering have also been proposed. This is an interesting and original article but it can be improved in order to be accepted for publication.

1) The electromechanical coupling strength gab=0.5gam used in the paper, and gam should be the coupling strength between the magnon and the microwave photons. My question is whether the electromechanical coupling strength is appropriate? It is advisable for author to cite relevant experimental articles for discussion.

2) The paper would gain a lot by connecting the work and real devices/experiments. Therefore, I would like the authors to comment on how realistic it is to build and measure this effect.

3) The notes in Figures 3-5 should be written in more detail to make it easier for the reader to quickly understand the information in the figure.

4) Hybrid magnon quantum systems are mentioned in the introduction section, and some interesting quantum phenomena related to magnon are also mentioned. Magnons have rich nonlinear properties, which should also be mentioned in the introduction so that readers can understand the field of magnonics more fully, such as the generation of optomagnonic frequency combs, magnonic frequency combs in cavity magnomechanics, magnon-induced high-order sideband generation, and so on.

5) Some formatting issues should also be noted, for example, the use of square brackets in line 94; in Figure 2-4, the image layout is suggested to be centered.

Reviewer 3 Report

Please see the following attachment.
